# SOAT1: A Suitable Target for Therapy in High-Grade Astrocytic Glioma?

**DOI:** 10.3390/ijms23073726

**Published:** 2022-03-28

**Authors:** Mario Löhr, Wolfgang Härtig, Almut Schulze, Matthias Kroiß, Silviu Sbiera, Constantin Lapa, Bianca Mages, Sabrina Strobel, Jennifer Elisabeth Hundt, Simone Bohnert, Stefan Kircher, Sudha Janaki-Raman, Camelia-Maria Monoranu

**Affiliations:** 1Department of Neurosurgery, University Hospital Wuerzburg, 97080 Wuerzburg, Germany; loehr_m1@ukw.de; 2Paul Flechsig Institute for Brain Research, University of Leipzig, 04103 Leipzig, Germany; wolfgang.haertig@medizin.uni-leipzig.de; 3Division of Tumor Metabolism and Microenvironment, German Cancer Research Center (DKFZ), 69120 Heidelberg, Germany; almut.schulze@dkfz-heidelberg.de; 4Department of Internal Medicine IV, University Hospital Munich, Ludwig-Maximilians-Universität Munich, 80336 Munich, Germany; matthias.kroiss@med.uni-muenchen.de; 5Department of Internal Medicine I, Division of Endocrinology and Diabetes, University Hospital, University of Wuerzburg, 97080 Wuerzburg, Germany; sbiera_s@ukw.de; 6Nuclear Medicine, Medical Faculty, University of Augsburg, 86156 Augsburg, Germany; constantin.lapa@uk-augsburg.de; 7Institute for Anatomy, University of Leipzig, 04103 Leipzig, Germany; bianca.mages@medizin.uni-leipzig.de; 8Institute of Pathology, Department of Neuropathology, University of Wuerzburg, Josef-Schneider-Str. 2, 97080 Wuerzburg, Germany; sabrina.strobel@uni-wuerzburg.de (S.S.); stefan.kircher@uni-wuerzburg.de (S.K.); 9Luebeck Institute of Experimental Dermatology, University of Luebeck, 23538 Luebeck, Germany; jennifer.hundt@uni-luebeck.de; 10Institute of Forensic Medicine, University of Wuerzburg, 97080 Wuerzburg, Germany; simone.bohnert@uni-wuerzburg.de; 11Donald B. and Catherine C. Marron Cancer Metabolism Center, Sloan Kettering Institute, Memorial Sloan Kettering Cancer Center, New York, NY 10065, USA; janakirs@mskcc.org

**Keywords:** SOAT1, glioblastoma, astrocytoma, *IDH1*/*2*, lipid droplets, mitotane, targeted therapy

## Abstract

Targeting molecular alterations as an effective treatment for isocitrate dehydrogenase-wildtype glioblastoma (GBM) patients has not yet been established. Sterol-O-Acyl Transferase 1 (SOAT1), a key enzyme in the conversion of endoplasmic reticulum cholesterol to esters for storage in lipid droplets (LD), serves as a target for the orphan drug mitotane to treat adrenocortical carcinoma. Inhibition of SOAT1 also suppresses GBM growth. Here, we refined SOAT1-expression in GBM and *IDH*-mutant astrocytoma, CNS WHO grade 4 (HGA), and assessed the distribution of LD in these tumors. Twenty-seven GBM and three HGA specimens were evaluated by multiple GFAP, Iba1, IDH1 R132H, and SOAT1 immunofluorescence labeling as well as Oil Red O staining. To a small extent SOAT1 was expressed by tumor cells in both tumor entities. In contrast, strong expression was observed in glioma-associated macrophages. Triple immunofluorescence labeling revealed, for the first time, evidence for SOAT1 colocalization with Iba1 and IDH1 R132H, respectively. Furthermore, a notable difference in the amount of LD between GBM and HGA was observed. Therefore, SOAT1 suppression might be a therapeutic option to target GBM and HGA growth and invasiveness. In addition, the high expression in cells related to neuroinflammation could be beneficial for a concomitant suppression of protumoral microglia/macrophages.

## 1. Introduction

Personalized therapies have been successfully developed during the past two decades for a subset of malignant tumor entities. However, for *isocitrate dehydrogenase (IDH)*-wildtype glioblastoma (GBM) patients, molecular targeted treatment has not yet been established, and the overall prognosis of patients with this highly malignant brain tumor remains poor. Emerging evidence demonstrates the important role of lipid metabolism in cancer cells. Identifying key aspects of the lipid metabolism that are specifically engaged with tumorigenesis suggests a new strategy to treat malignancies [1,2]. Recently, increased lipid metabolism, regulated by the transcription factor sterol regulatory element-binding protein-1 (SREBP-1), has been shown to be characteristic for GBM. SREBP- 1 in its inactive state is an integral membrane protein of the endoplasmic reticulum. Sterol-O-Acyl Transferase 1 (SOAT1) is one of the key target enzymes of SREBP-1 activation and catalyzes the esterification of free cholesterol with fatty acids to cholesterol esters, which are then transferred into lipid droplets (LD) as a storage pool for cholesterol and fatty acids.

SOAT1 has been shown to be of relevance as a prognostic marker and potential therapeutic target for several tumors. Thus, high SOAT1 expression has been demonstrated to be associated with unfavorable prognosis in high-risk prostate cancer (PCa) [3]. Inhibition of cholesteryl ester formation with avasimibe, a SOAT1 inhibitor initially developed for the treatment of atherosclerosis [4], in a cell culture model of PCa, was able to reduce the viability of the cells and to lower in vitro indicators of cell migration and invasiveness [5]. Generally, inhibition of SOAT1 could effectively suppress SREBP-1 and, consequently, GBM growth [6,7]. In particular, avasimibe suppresses GBM cell growth in vitro significantly [8,9]. Mitotane is the only FDA-approved SOAT1 inhibitor and is in clinical use for the treatment of the orphan disease adrenocortical carcinoma [10]. SOAT1 expression is associated with unfavorable prognosis but does not predict response to mitotane monotherapy. Patients treated with mitotane frequently exhibit moderate to severe neurological adverse effects, such as dizziness and fatigue, which are not yet mechanistically understood [11].

Higher SOAT1 expression has been demonstrated in GBM compared to astrocytoma of lower malignancy (CNS WHO grades 2–3), with absence of expression in pilocytic astrocytoma, CNS WHO grade 1, and control brains with cortical dysplasia [3]. The same study reported that the amount of cytoplasmic LD is elevated in GBM and inversely correlates with patients’ survival [3]. These results have to be taken with caution due to the high tumor heterogeneity of GBM with a high proportion of glioma-associated microglia/macrophages.

The aim of our study was to refine the analysis of SOAT1 expression in the GBM microenvironment. We therefore assessed SOAT1 expression by immunohistochemistry in tumor tissue from patients with GBM and *IDH*-mutant astrocytoma, CNS WHO grade 4 (HGA), using normal brain as a control that could also be relevant for the observed neurotoxic side effects of SOAT1 inhibitors such as mitotane.

## 2. Results

### 2.1. Tissue Samples

We retrospectively evaluated specimens from 27 *IDH* wildtype GBM, CNS WHO grade 4 and 3 HGA, CNS WHO grade 4 with immunohistochemical evidence of the IDH1 R132H mutation. All patients were resected or biopsied at the Department of Neurosurgery of the University Hospital Würzburg, Germany, between January 2012 and March 2016. The tumors were histologically assessed and graded on formalin-fixed and paraffin embedded tissue sections by experienced neuropathologists, according to the criteria of the World Health Organization [12]. (Table 1).

### 2.2. Single Staining

Immunoperoxidase single staining of GBM samples suggested SOAT1 to be more pronounced in microglia and macrophages rather than in tumor cells. An unequivocal expression of SOAT1 in tumor cells could not be definitely established (Figure 1).

In normal brain, no specific SOAT1 expression was observed in neurons, oligodendrocytes, and astrocytes in any of the analyzed regions. However, the microglia, cells of the choroid plexus and circulating intravascular monocytes showed specific staining (Figure 2). In macrophages of peripheral organs (liver, lung, tonsil, and lymph node) SOAT1 was also strong expressed. Figure 3 shows an example of the staining in lung and tonsil.

### 2.3. Multiple Fluorescence Labeling

Multiple carbocyanine labeling revealed SOAT1 colocalization with either astroglial GFAP or Iba1 in microglia/macrophages (Figure 4), additional to its colocalization with IDH1 R132H in HGA cells. (Figure 5). In all tumor samples, the proportion of SOAT1-positive microglia/macrophages was higher than that of tumor cells.

### 2.4. Oil Red O Staining

Notably, GBM and HGA tumor samples showed differently distributed LD. Whereas GBM exhibited abundant LD in tumor cells, HGA appeared nearly devoid of them (Figure 6).

## 3. Discussion

Despite aggressive treatment with surgery, radiation, and chemotherapy, GBM remains an incurable and invariably recurrent brain tumor. To date, no pharmacological intervention has been demonstrated to substantially influence the course of the disease. For this reason, there is increasing interest in the development of targeted therapies not only aiming at the tumor cells but also at the microenvironment, including the macrophage/microglia compartment, which is known to be crucial for tumor invasiveness and progression.

In this study, we analyzed the expression of SOAT1, recently identified as the target molecule of mitotane, which is approved as an orphan drug for the treatment of adrenocortical carcinoma. Our goal was to explore its expression in GBM and HGA, the most frequent malignant brain tumors in adults [13,14], as a rationale for its future therapeutic inhibition in these tumors.

In our panel of 27 GBM and 3 HGA, CNS WHO grade 4, only a small proportion of the tumor cells displayed SOAT1-immunoreactivity. In contrast, strong and extensive expression was observed in glioma-associated macrophages, in both tumor entities. Geng et al. described a higher expression in GBM tissue compared to astrocytoma of lower histological grades (2–3) based on single immunolabeling [3]. However, this approach might prevent the high precision in identification of positive and negative cell types within the tumor tissue. GBM contain a large proportion of macrophages compared to astrocytomas of lower grades, which predominantly comprise small-sized ameboid microglia [15]. This could be an explanation for the higher SOAT1-expression in GBM compared to low-grade astrocytoma in the mentioned study. In recent years, there has been more and more discussion about the modulation of the immune cells as a therapeutic approach in GBM. The high SOAT1-expression in tumor-associated macrophages could be the basis for a therapeutic attempt with mitotane, in GBM patients with no further therapeutic options. Future investigations of the in vitro effect of SOAT1 inhibition on macrophage polarization would be of interest in this regard.

By applying triple immunofluorescence labeling, we are the first to provide evidence for SOAT1 co-expression with immunoreactivities for Iba1 and IDH1 R132H positive cells, respectively.

Another interesting and novel result of the present study was the notable difference in the amount of cytoplasmic LD between GBM and HGA. As already mentioned, elevated lipogenesis, regulated by SREBP-1, is a novel characteristic of GBM. SREBP-1 activation is negatively regulated by endoplasmic reticulum cholesterol, and SOAT1 is a key enzyme converting endoplasmic reticulum cholesterol to esters for storage in LD. In a previously published study, an inverse correlation between the amount of LD and patient survival was observed in GBM, however, without *IDH*-status specification [3]. In our cohort, the HGA were nearly completely devoid of LD accumulation, whereas GBM showed an abundance of them. This might be a consequence of the differences in pathogenesis between both tumor entities. Tumor evolution studies involving the sequencing of paired initial and recurrent *IDH*-mutant tumors have suggested that mutation of the *IDH* gene is an early event in tumor formation [16,17], and the mutation indirectly alters the level of lipid synthesis [18,19]. The results of our analysis underline the differences between both tumor subtypes and raise doubt regarding a potential therapy success by suppression of SOAT1 in HGA.

Studies regarding SOAT1 expression in healthy brains were lacking to date. In our study, the expression of SOAT1 appeared restricted to microglial cells, whereas other cell types, such as neurons, astrocytes, and oligodendroglia, remained negative. This suggests that the neurological adverse events seen in mitotane-treated patients are not on-target effects mediated by SOAT1 inhibition. This is in accordance with several studies reporting moderate neurotoxicity in individual patients treated with high-dose mitotane for adrenocortical carcinoma [20,21,22]. However, we could demonstrate SOAT1 expression in peripheral macrophages. The clinical impact of this finding remains unclear regarding a possible relationship to the well-known systemic side effects of mitotane.

## 4. Materials and Methods

Formalin-fixed normal brain tissue obtained from the local Brain Bank served as controls. Two whole brains from patients without brain tumor or other cerebral lesions were cut into coronal slices. Following a standardized protocol, 17 brain specimens were sampled in each case, including areas such as the frontal, temporal, parietal, occipital lobes, the cingulate gyrus and the striatum, the basal forebrain including the amygdala, thalamus, and the anterior and posterior hippocampus; midbrain including the substantia nigra; pons including the locus coeruleus; medulla oblongata, vermis, and cerebellar cortex. We also analyzed the SOAT1 expression in macrophages of peripheral organs (liver, lung, tonsil, and lymph node).

All tissue samples were obtained with the consent of the patients or next of kin and according to the guidelines of the national and local ethics committees. The study was approved by the local ethics committee of the University of Würzburg (internal application number 99/11) and performed in accordance to the ethical standards described in the most recent version of the Declaration of Helsinki.

### 4.1. Single Immunohistoperoxidase Staining

Sequential 3 µm-thick paraffin sections were stained by applying classical immunohistochemical methodology. Sections were deparaffinized by drying on Superfrost plus slides (Fisher Scientific, Schwerte, Germany), heated at 56 °C overnight, and washed with mixed xylenes, 100% ethanol, and 95% ethanol.

The astroglial origin of astrocytic tumor cells was confirmed by immunopositivity for the glial fibrillary acidic protein (GFAP; 1:200, mouse monoclonal antibody, Clone 6F2, Dako, Hamburg, Germany). The astrocytes of adjacent brain parenchyma served as internal positive controls.

A monoclonal mouse antibody directed against CD68 (1:200, clone IS609, Dako, Hamburg, Germany) was applied to identify intratumoral microglial cells and macrophages.

*IDH1* mutational status was determined utilizing a specific antibody for the R132H mutation (1:100, monoclonal mouse antibody, clone H09, Dianova, Hamburg, Germany). Cases with a mutation confirmed by sequence analysis and immunohistochemistry) served as controls. In case of immunohistochemical negativity, the genomic DNA was extracted from the tumor tissue using a DNA Isolation Kit for formalin-fixed and paraffin-embedded tissue (Qiagen, Hilden, Germany) and the region around codon 132 of IDH1 and codon 172 of IDH2 was amplified by PCR using specific primers (Life Technologies, Darmstadt, Germany). The purified amplificates were analyzed by pyrosequencing.

Immunoperoxidase labeling of SOAT1 was performed as previously described based on a rabbit polyclonal antibody (1:1000, rabbit polyclonal antibody, ab39327, Abcam, Cambridge, UK) [10]. Adrenal gland tissue served as the positive control for this staining.

For detection, link- and label-antibody from the SS Multilink HRP kit (DCS, LP000-UL, Hamburg, Germany) and the ultraView Universal DAB Detection Kit (Ventana Medical Systems, 760-500, Darmstadt, Germany) were used according to the manufacturer’s instructions.

All immunoperoxidase-labeled sections were counterstained for 2 min with hematoxylin (Sigma-Aldrich, Taufkirchen, Germany).

### 4.2. Oil Red O Staining

To detect lipid droplets in tumor samples, fresh-frozen GBM and HGA tissue from our study cohort was stained with lipid stain Oil Red O (Dianova, Hamburg, Germany) according to the manufacturer’s protocol. Four µm-thick frozen tissue sections were incubated in propylene glycol followed by incubation in Oil Red O solution and differentiation in propylene glycol (Sigma-Aldrich, Taufkirchen, Germany). After incubation in hematoxylin, the slides were rinsed with water and finally covered using an aqueous mounting medium (ab64230, Abcam, Cambridge, UK).

### 4.3. Immunofluorescence Labeling

Triple immunofluorescence staining was performed in order to identify the SOAT1 positive cell types in the GBM and HGA samples. Briefly, slides mounted with deparaffinized 5 µm thick sections were extensively washed with 0.1 M Tris-buffered saline, pH 7.4 (TBS), prior to blocking nonspecific binding sites for subsequently applied immunoreagents with 5% normal donkey serum in TBS containing 0.3% Triton X-100 for 1 h in a humidity chamber. The tissue was incubated overnight with one of the following mixtures, which all contained rabbit-anti-SOAT1 antibody (1:100 in the blocking solution, abcam, Cambridge, UK): I) guinea pig-anti-GFAP (1:200; 173,004, Synaptic Systems, Göttingen, Germany) and biotinylated *Solanum tuberosum* lectin (STL; 20 µg/mL; B-1165, Vector, Burlingame, CA, USA); II) guinea pig-ionized calcium binding adapter molecule-1 (Iba1; 1:100; 234,004, Synaptic Systems, Göttingen, Germany) and biotinylated STL (20 mg/mL; Vector); or III) mouse-anti-IDH (1:20; Dianova, Hamburg, Germany) and guinea-pig-anti-Iba1 (1:100; Synaptic Systems, Göttingen, Germany). Following several rinses with TBS, the sections were left to react for 1 h with mixtures of carbocyanine (Cy)3-donkey-anti-rabbit IgG, Cy2-donkey-anti-guinea pig, and Cy5-streptavidin (for I and II) or Cy3-donkey-anti-rabbit IgG, Cy2-donkey-anti-mouse IgG, and Cy5-donkey-anti-guinea pig IgG (for III); all fluorochromated antibodies were from Dianova as supplier for Jackson ImmunoResearch West, Grove, PA, USA and used for 1 h at 20 µg/mL TBS containing 2% bovine serum albumin. Next, the tissue was washed again with TBS, and its autofluorescence was quenched by treatment with Sudan Black B according to Schnell et al. (1999). Finally, the sections were coverslipped with glycerol gelatin (GG1, Sigma-Aldrich, Taufkirchen, Germany).

In the histological control experiments, the omission of primary antibodies and biotinylated STL resulted in the expected absence of any cellular staining.

Pictures from multiple fluorescence labeling were made with a microscope Biorevo BZ-9000 (Keyence, Neu-Isenburg, Germany).

## 5. Conclusions

SOAT1 suppression might be a new therapeutic option in regard to targeting GBM growth and invasiveness. The higher expression in cells related to neuroinflammation compared to the tumor cells, could be of significance for a concomitant suppression of protumoral microglia/macrophages. The importance of the newly reported SOAT1 expression in peripheral organs remains largely unclear and requires further investigation.

## Figures and Tables

**Figure 1 ijms-23-03726-f001:**
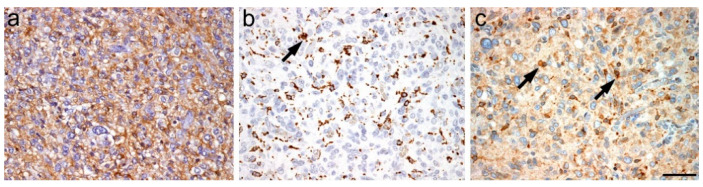
(**a**) GFAP staining of glioblastoma tumor cells; (**b**) CD68 positivity of tumor associated microglia and macrophages; (**c**) SOAT1 expression in GBM (scale bar 200 µm).

**Figure 2 ijms-23-03726-f002:**
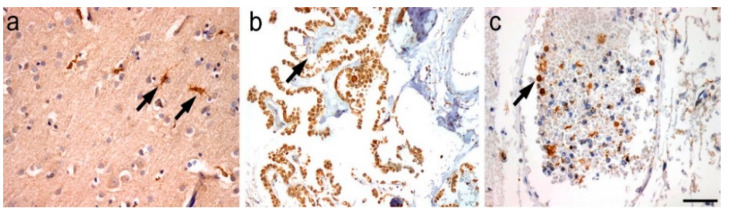
(**a**) SOAT1 expression in microglial cells of normal brain (frontal cortex), (**b**) in cuboidal cells of the choroid plexus, and (**c**) in circulating monocytes (scale bar 200 µm).

**Figure 3 ijms-23-03726-f003:**
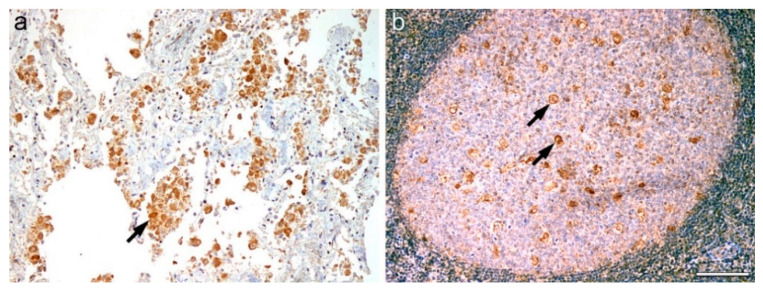
(**a**) SOAT1 expression in alveolar macrophages of lung tissue and (**b**) in the germinal center histiocytes of tonsillar tissue (scale bar 200 µm).

**Figure 4 ijms-23-03726-f004:**
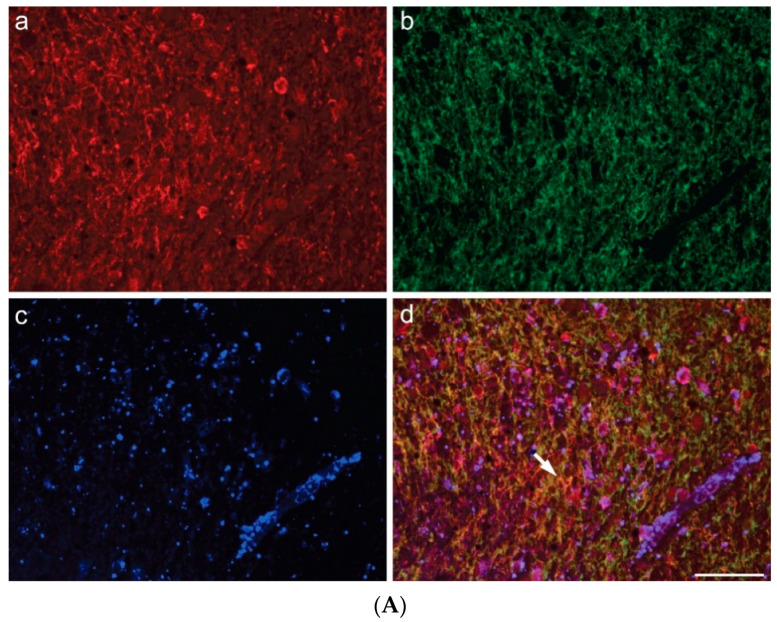
(**A**) Coexpression of SOAT1 and GFAP in glioblastoma: (**a**) SOAT1; (**b**) GFAP; (**c**) *Solanum tuberosum* lectin (STL) staining of vessels and microglia/macrophages; (**d**) merge (scale bar 200 µm). (**B**) Coexpression of SOAT1 and Iba1 in glioblastoma (arrows): (**a**) SOAT1; (**b**) Iba1; (**c**) STL; (**d**) merge (scale bar 100 µm).

**Figure 5 ijms-23-03726-f005:**
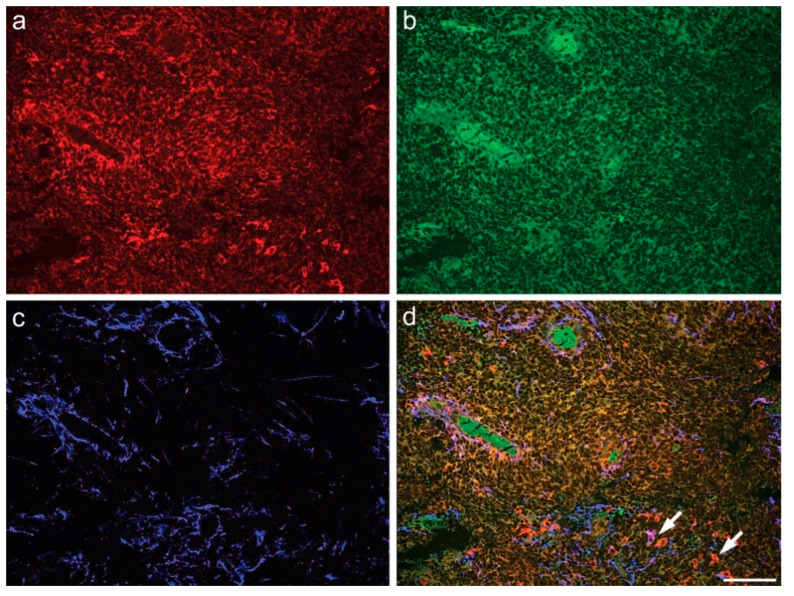
Coexpression of IDH1R132H and SOAT1 in an IDH1-mutated astrocytoma, CNS WHO grade 4 (arrows): (**a**) SOAT1; (**b**) IDH1 R132H; (**c**) *Solanum tuberosum* lectin (STL), predominantly revealing endothelia; (**d**) merge (scale bar 200 µm).

**Figure 6 ijms-23-03726-f006:**
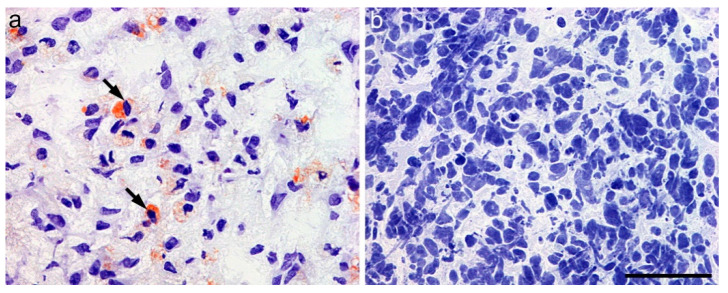
Intracytoplasmic lipid droplets in (**a**) glioblastoma and (**b**) astrocytoma, IDH-mutant, CNS WHO grade 4 (scale bar 100 µm).

**Table 1 ijms-23-03726-t001:** Demographic and molecular data of glioma samples.

No.	Age at Surgery	Gender	Diagnosis	WHO	MGMT-Status	*IDH*-Status
1	79	M	GBM	4	methylated	wt
2	63	M	GBM	4	unmethylated	wt
3	72	F	GBM	4	methylated	wt
4	80	M	GBM	4	unmethylated	wt
5	67	F	GBM	4	methylated	wt
6	74	F	GBM	4	methylated	wt
7	68	M	GBM	4	unmethylated	wt
8	58	M	GBM	4	methylated	wt
9	77	M	GBM	4	methylated	wt
10	65	F	GBM	4	methylated	wt
11	71	M	GBM	4	methylated	wt
12	66	F	GBM	4	unmethylated	wt
13	78	M	GBM	4	methylated	wt
14	63	M	GBM	4	methylated	wt
15	57	F	GBM	4	methylated	wt
16	61	F	GBM	4	methylated	wt
17	69	M	GBM	4	unmethylated	wt
18	73	M	GBM	4	methylated	wt
19	56	F	GBM	4	unmethylated	wt
20	62	F	GBM	4	methylated	wt
21	54	M	GBM	4	methylated	wt
22	56	M	GBM	4	unmethylated	wt
23	81	F	GBM	4	methylated	wt
24	83	F	GBM	4	unmethylated	wt
25	72	M	GBM	4	methylated	wt
26	71	M	GBM	4	methylated	wt
27	59	M	GBM	4	methylated	wt
28	41	F	HGA	4	methylated	mutant
29	39	M	HGA	4	methylated	mutant
30	47	M	HGA	4	methylated	mutant

GBM: glioblastoma; HGA: high-grade astrocytoma; WHO: CNS WHO grade; MGMT-Status: O^6^-methylguanine-DNA methyl-transferase promoter methylation status; *IDH*-status: isocitrate dehydrogenase status; wt: wildtype.

## Data Availability

The study did not report any data.

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
