# Peer review of "SOAT1: A Suitable Target for Therapy in High-Grade Astrocytic Glioma?"

_ijms, 2022, doi:10.3390/ijms23073726_

Round 1

Reviewer 1 Report

The authors demonstrated that SOAT1 is related to the tumor environment of GBM, especially macrophages. This article is interesting and might propose a novel therapeutic target against GBM. However, the reviewer requests the following issues for further review:

  1. The authors mentioned that SOAT1 might not be directly related to GBM. But the evidence is weak. The authors had better demonstrate the correlation between the prognosis of patients with GBM (especially macrophages in the tumor tissues) and SOAT1 expression. If the authors could not examine their own data, it is necessary to examine it using public databases.
  2. The authors discussed the neurotoxicity of mitotane as a therapeutic agent against adrenocortical carcinoma. However, the relationship or rationale with GBM had not been described clearly. The authors had better mention it more precisely and more clearly. Moreover, the authors had better change the title of this article.

Reviewer 2 Report

The title does not reflect the content of the article. In this article, the authors investigated SOAT1 expression in GBM and not patient survival.

The authors studied 27 GBM tumors. The authors also studied 3 grade 4 brain tumors with an IDH mutation. These 3 tumors are not classified. Authors must write whether it is GBM with an IDH mutation or another type of cancer.

GFAP is not a marker of GBM cancer cells. Often, in GBM, tumor expression of GFAP is lowered. The authors must examine SOAT1 expression in the context of the GBM cells marker. Most primary GBMs do not have the IDH mutation.

Wilhelmsson U, Eliasson C, Bjerkvig R, Pekny M. Loss of GFAP expression in high-grade astrocytomas does not contribute to tumor development or progression. Oncogene. 2003;22(22):3407-11. doi: 10.1038/sj.onc.1206372.

The authors need to develop the importance of SOAT1 in macrophages / microglia for in vitro studies. They should investigate the effect of a SOAT1 inhibitor on macrophage polarization and the expression of certain genes (VEGF, COX-2, ...) in macrophages.

Round 2

Reviewer 1 Report

This article has been improved, therefore, this is acceptable in our journal.

However, the reviewer wants the authors to submit the next paper about the correlation between prognosis and SOAT1 expression and the development of a novel therapeutic strategy against GBM.